# Preparation, Characterization and Permeation Study of Topical Gel Loaded with Transfersomes Containing Asiatic Acid

**DOI:** 10.3390/molecules27154865

**Published:** 2022-07-29

**Authors:** Shakthi Apsara Thejani Opatha, Varin Titapiwatanakun, Korawinwich Boonpisutiinant, Romchat Chutoprapat

**Affiliations:** 1Pharmaceutical Sciences and Technology Program, Faculty of Pharmaceutical Sciences, Chulalongkorn University, Bangkok 10330, Thailand; 6272015933@student.chula.ac.th (S.A.T.O.); varin.t@pharm.chula.ac.th (V.T.); 2Department of Pharmaceutics and Industrial Pharmacy, Faculty of Pharmaceutical Sciences, Chulalongkorn University, Bangkok 10300, Thailand; 3Innovative Natural Products from Thai Wisdoms (INPTW), Faculty of Integrative Medicine, Rajamangala University of Technology Thanyaburi, Pathumthani 12130, Thailand; korawinwich_b@rmutt.ac.th

**Keywords:** asiatic acid, in vitro permeation, transfersomal gel, nanocarriers

## Abstract

The objective of this study is to investigate the in vitro permeation of asiatic acid (AA) in the form of a topical gel after entrapment in transfersomes by Franz diffusion cells. Transfersomes composed of soybean lecithin and three different edge activators including Tween 80 (TW80), Span 80 (SP80) and sodium deoxycholate (SDC) at the ratio of 50:50, 90:10 and 90:10, respectively, together with 0.3% *w*/*w* of AA, were prepared by a high-pressure homogenization technique and further incorporated in gels (TW80AATG, SP80AATG and SDCAATG). All transfersomal gels were characterized for their AA contents, dynamic viscosity, pH and homogeneity. Results revealed that the AA content, dynamic viscosity and pH of the prepared transfersomal gels ranged from 0.272 ± 0.006 to 0.280 ± 0.005% *w*/*w*, 812.21 ± 20.22 to 1222.76 ± 131.99 Pa.s and 5.94 ± 0.03 to 7.53 ± 0.03, respectively. TW80AATG gave the highest percentage of AA penetration and flux into the Strat-M^®^ membrane at 8 h (8.53 ± 1.42% and 0.024 ± 0.008 mg/cm^2^/h, respectively) compared to SP80AATG (8.00 ± 1.70% and 0.019 ± 0.010 mg/cm^2^/h, respectively), SDCAATG (4.80 ± 0.50% and 0.014 ± 0.004 mg/cm^2^/h, respectively), non-transfersomal gels (0.73 ± 0.44 to 3.13 ± 0.46% and 0.002 ± 0.001 to 0.010 ± 0.002 mg/cm^2^/h, respectively) and hydroethanolic AA solution in gel (1.18 ± 0.76% and 0.004 ± 0.003 mg/cm^2^/h, respectively). These findings indicate that the TW80AATG might serve as a lead formulation for further development toward scar prevention and many types of skin disorders.

## 1. Introduction

Asiatic acid (AA) is a naturally occurring pentacyclic triterpenoid that can possess many biological activities such as anti-inflammatory, scar prevention, anti-skin cancer and anti-microbial ones, as well as significant antioxidant activities [1,2,3,4,5,6,7,8]. According to the literature, AA significantly inhibited the intracellular concentrations of reactive oxygen species (ROS) and moderately inhibited the multiple inflammatory pathways such as the Akt pathway and transcription factor nuclear factor kappa-B (NF-κB) signaling pathway leading to a normal wound healing process and scar prevention [2,5,9]. Hence, this study focuses on the utilization of the potential therapeutic activities of AA toward scar prevention in a much efficacious way by efficient dermal delivery of AA using transfersomal gel formulations. However, AA is a highly lipophilic molecule with a log *p*-value of 5.7; therefore, it has shown a significantly low skin permeability [3]. A new vesicular delivery system called “Transfersomes” has been identified as one of the major advancements in vesicle research [10,11,12,13] that possesses the ability of deeper skin permeation and even reaches to blood circulation without compromising vesicle integrity [11,14,15]. Transfersomes have been examined and identified to have the capability to deliver active agents through the skin barrier (stratum corneum) and thereby improve the drug permeation [16,17].

Transfersomes are composed of amphipathic ingredients (such as phosphatidylcholine), which self-assemble into a lipid bilayer in an aqueous solvent and close into a simple lipid vesicle, and a bilayer softening component that also known as the edge activator (EA) (such as a biocompatible surfactant), which greatly increases the lipid bilayer flexibility and permeability [18]. Due to the self-optimizing and ultra-deformable nature of transfersomes, they can easily deform and squeeze through narrow constrictions of the skin that are significantly smaller than the size of the vesicles [19,20,21]. This unique structure and composition facilitate the entrapment of hydrophilic, lipophilic and amphiphilic drugs with relatively high permeation efficiency, as well as controlled and potentially targeted drug delivery [11]. In our previous study, asiatic acid-loaded nano-transfersomes (AAT) composed of soybean lecithin and three different edge activators including TW80, SP80 and SDC were developed with the primary focus on the effects of edge activators on the physicochemical characteristics, in vitro permeation and anti-inflammatory activity of AAT. It was found that transfersomes significantly increased the percentage of AA penetration and flux into the Strat-M^®^ membrane and anti-inflammatory activity of AA [22]. However, most of the vesicular drug delivery systems (such as liposomes, niosomes and transfersomes) possess a very low viscosity that makes them unable to retain at the site of application for a long time and makes their topical application inconvenient [23,24]. Transfersome-based gels, so-called transfersomal gels, could facilitate the adhesion to the skin and the controlled release of the transfersomes, and thereby a targeted controlled release of bioactive agents entrapped in the vesicles could be achieved. This is often required to achieve the desired therapeutic effect [25]. The pH of a gel formulation is another important parameter that needs to be considered during formulation development for dermal application. Human skin pH was reported to be between 4.5 and 5.5 [26]. According to previous studies related to transfersomal gel formulations, the skin compatible pH range of the prepared formulations were reported as pH 5.3 to 7.6 [27,28,29,30]. In the present study, a gel formulation for dermal administration of asiatic acid-loaded transfersomes (AATG) has been developed in an attempt to overcome the mentioned drawback of transfersomes and provide a therapeutic gel for skin disorders. The transfersomal formulations are mixed with the gelling agents (hydroxyethyl acrylate/sodium acryloyldimethyl taurate copolymer and polyacrylate crosspolymer-6), silicone-based emulsifier, hyaluronic acid and preservative to obtain the desired transfersomal gels. The pH values of the prepared transfersomal gels are expected to be within acceptable range.

Moreover, the efficient dermal delivery of asiatic acid using transfersomes in topical gel formulation has never been reported. Hence, the focus of the present study is to determine the permeability across the hypothetical skin of AA after incorporation in the transfersomal gels and other important parameters such as pH, dynamic viscosity, appearance, and homogeneity of the transfersomal gel formulations in order to identify the promising formulation with good stability characteristics for further in vivo studies.

## 2. Results and Discussion

According to our previous study, the prepared blank transfersomes composed of soybean lecithin and three different edge activators including TW80, SP80 and SDC at the ratio of 50:50, 90:10 and 90:10, respectively, which gave the highest physical stability, were selected to entrap 0.3% AA (TW80AAT, SP80AAT and SDCAAT). The prepared TW80AAT, SP80AAT and SDCAAT had vesicular sizes of 63.54 ± 2.51, 45.71 ± 2.03 and 27.15 ± 0.95 nm, respectively. The TW80AAT exhibited the highest percentage of EE (90.84 ± 2.99%), followed by the SP80AAT (87.25 ± 3.24%) and SDCAAT (79.87 ± 7.21%) [22]. However, asiatic acid-loaded transfersomes (AAT) formulations are not convenient for topical application because of their poor adhesion and the difficulty of achieving a site-specific application caused by the fluidic nature of the formulations [31]. Gels are well-known for their convenience in usage, better application, adhesion, distribution and drug permeation through the transdermal routes caused by their hydrophilic nature [32]. Therefore, the obtaining AAT formulations were further incorporated into a gel base (AATG) and used for physicochemical property evaluation and in vitro skin permeation study.

As previously mentioned, most of the vesicular drug delivery systems (such as liposomes, niosomes and transfersomes) possess a very low viscosity that makes them unable to retain at the site of application for a long time and makes their topical application inconvenient [23,24]. Therefore, in the present study, the gel was developed to facilitate the adhesion to the skin and the controlled release of the transfersomes, as well as to provide a therapeutic gel for skin disorders. However, being a topical treatment for skin disorder, the aesthetics of the formulation was considered as one of the important factors to improve patient adherence to the treatment [33]. Therefore, the gel formulation was carefully developed using selected components not only to improve skin adhesion, but also to improve its aesthetic property. The functions of each component used in the formulation are described as follows:

Hydroxyethyl acrylate/sodium acryloyldimethyl taurate copolymer is a pre-neutralized synthetic polymer with a good resistance property to electrolytes in a wide range of pH 3–12. It was used in the formulations as a gelling agent [34]. Polyacrylate crosspolymer-6 is also pre-neutralized synthetic polymer acting as a gelling and stabilizing agent. This component could deposit over the outer layers of the skin because of its high molecular weight and thereby facilitate the formation of a barrier to prevent water loss and hydrate the skin [35]. The silicone-based emulsifier (dimethicone-based emulsifier) was used in the formulations as a texture and sensory modifier. It could be incorporated in water-based products to provide a shiny, light and silky texture, as well as ease of spreading, to the formulation [36,37]. Last, but not least, hyaluronic acid and phenoxyethanol were used in the formulations as the moisturizing component and preservative, respectively.

### 2.1. Evaluation of AA Transfersomal Gels (AATGs)

#### 2.1.1. Appearance 

All prepared transfersomal gel formulations appeared to be white in color and opaque. The gel formulation containing asiatic acid-loaded transfersomes composed of soybean lecithin with SDC (SDCAATG) exhibited a slight translucent appearance compared to those containing soybean lecithin with TW80 (TW80AATG) and soybean lecithin with SP80 (SP80AATG).

#### 2.1.2. Viscosity

The dynamic viscosities of the TW80AATG, SP80AATG and SDCAATG are shown in Table 1. The highest dynamic viscosity obtained from SP80AATG could be imparted because of the hydrophobicity of SP80 incorporated in the gel formulation [25,38]. The TW80AATG and SDCAATG exhibited a pseudoplastic flow behavior (Figure 1A,C), which means that their viscosities decrease with an increase in shear rate [27,39]. According to Gupta et al. (2012), pseudoplasticity or shear thinning of gel could be due to its colloidal network structure, which aligns itself in the direction of the shear and thereby decreases in the apparent viscosity when the shear rate increases [17]. Moreover, pseudoplastic flow behavior displayed by the transfersomal gels could facilitate the spreadability of gels once shear stress is applied, which ensures the maximum area of coverage during gel application, whereas during static conditions, the gels are capable of returning to the viscous gel form. Therefore, the developed TW80AATG and SDCAATG achieved the desirable and ideal requirement of the topical gels [39]. The SP80AATG at 4 °C exhibited an elastic deformation till the shear stress point 200 s^−1^ and was followed by gel fracture in a brittle manner indicating an abrupt stress decay (Figure 1B). However, at 25 °C and 40 °C, SP80AATG showed a broader and smooth strain evolution in the yielding zone as shown on the graph, which indicated a more ductile fracture with considerable plastic deformation or permanent deformation [40].

#### 2.1.3. pH

As shown in Table 1, the pH values of TW80AATG, SP80AATG and SDCAATG were 6.06 ± 0.09, 5.94 ± 0.03 and 7.53 ± 0.03, respectively. The blank gel had pH value of 6.37 ± 0.03. The slight increase in the pH was observed for SDCAATG. This could be due to the basic pH of SDC edge activator (around pH 8.4). Previous studies had demonstrated that gel formulations with a pH in the range of 5.3 to 7.6 was compatible with human skin [27,28,29,30]. Therefore, the pH values obtained from all transfersomal gels were found to be acceptable as the formulation is used for dermal application.

#### 2.1.4. AA Content

The AA content in the TW80AATG, SP80AATG and SDCAATG were 0.280 ± 0.005, 0.279 ± 0.001 and 0.272 ± 0.006 g per 100 g of gel, respectively. Since the expected AA content was 0.286 g per 100 g of gel (Table 1), therefore the percentages of AA in all prepared gel formulations were found to be varied from 95 to 98% of expected content. This result indicated that the drug loss during the formulation development was insignificant. 

#### 2.1.5. Homogeneity

The developed transfersomal gels exhibited a pleasant and good homogeneous appearance. There were no gritty particles and no phase separation observed in all three gel formulations. Dermatix Ultra™, a commercially available scar-relieving gel was used as the control. Small quantities of transfersomal gel and Dermatix gel (separately) were pressed between the thumb and the index finger. The consistency of the gels was noticed. The homogeneity was detected by rubbing a small quantity of the gel on the skin of the back of the hand. The grittiness of the transfersomal gels was also observed in the same way. Both control and transfersomal gels exhibited good homogeneity, uniform consistency and no grittiness and showed an absence of any lumps. Therefore, according to the homogeneity grading system, all three gels were graded as “good”. The control gel was identified as between good and fair since the applicants felt the texture of that gel as oily, compared to the transfersomal gels. 

#### 2.1.6. In Vitro Permeation 

The in vitro permeation studies were conducted for the freshly prepared AATGs and control gels using Franz diffusion cell with a total exposure period of 8 h to minimize the interference from the variation of physicochemical properties of the formulation, especially AA content, over time on the analysis. Figure 2 shows the physical appearance of freshly prepared AATGs (TW80AATG, SP80AATG and SDCAATG) and control formulations (physically mixed AA dispersions in gel; TW80AA, SP80AA, SDCAA and hydroethanolic solution of AA in gel) for the in vitro permeation study. The percent amount of AA penetrated into the Strat-M^®^ membrane and flux of AA into the Strat-M^®^ membrane over an 8 h period was plotted against the function of time to evaluate the release profiles (Figure 3).

At 0.5 h, the highest percentage of AA penetration (23.42 ± 1.10%) and the flux (1.055 ± 0.151 mg/cm^2^/h) of AA were demonstrated by SDCAATG, and the obtained values were significantly higher than those of other formulations (*p* < 0.05). The second highest percentage of AA penetration and fluxes of AA into the Strat-M^®^ membrane were exhibited by TW80AATG (9.24 ± 0.59% and 0.414 ± 0.084 mg/cm^2^/h, respectively), which were not significantly different from TW80AA dispersion in gel (7.51 ± 0.92% and 0.397 ± 0.045 mg/cm^2^/h, respectively) (*p* > 0.05). SP80AATG showed the significantly lowest percentage of AA penetration and flux (1.55 ± 0.62% and 0.062 ± 0.043 mg/cm^2^/h, respectively) compared to all other gel formulations (Figure 3). According to data obtained from HPLC analysis, the AA concentration in the receptor fluid was undetectable throughout the permeation study period for all tested formulations. Therefore, for the SDCAATG formulation, the highest amount of AA within the Strat-M^®^ membrane was achieved after 0.5 h, which was probably due to its relatively low dynamic viscosity. It was reported that the viscosity of formulations could significantly regulate and affect the drug permeation. The lower the viscosity of formulation, the faster the drug release and the higher the drug permeation [17]. Moreover, the smallest vesicle size of SDCAAT in gel may amplify its diffusion from the gel into the membrane [29,38]. Nevertheless, the high percentage of cumulative amount of AA permeated per unit area (mass unit/cm^2^) observed from the SDCAATG may lead to an early saturation of the membrane and subsequent increase of membrane surface pressure, which could drive drug molecules to move across the membrane against the concentration gradient, from lower concentration to higher concentration, this so-called reverse diffusion resulting in a decline of AA amount within the Strat-M^®^ over the rest of the test period [32,41,42].

At 4 h, the percentage of AA penetration and flux of TW80AATG (14.20 ± 0.47% and 0.079 ± 0.015 mg/cm^2^/h, respectively) were significantly higher than those of SDCAATG (10.31 ± 0.58% and 0.059 ± 0.012 mg/cm^2^/h, respectively), SP80AATG (7.68 ± 1.41% and 0.032 ± 0.007 mg/cm^2^/h, respectively) and controls (*p* < 0.05). The highest penetration of AA obtained from TW80AATG after 4 h was possible due to the presence of relatively high amount of edge activator (TW80) in the formulation (lipid:edge activator; 50:50) that enhanced deformation of TW80AAT leading to increased permeation of the vesicles and subsequent AA release. Moreover, the penetration-enhancing effects of transfersomes are associated with its self-optimizing and ultra-deformable nature. They can easily deform and squeeze through narrow constrictions of the skin that are significantly smaller than the size of the vesicles by following the natural osmotic gradient across the epidermis under a nonocclusive application [11]. Edge activators incorporated into the transfersome structure facilitate the destabilization of the vesicle’s lipid bilayer, enhancing its fluidity and elasticity, and minimizing the risk toward vesicle ruptures in the skin [18,43]. A relatively high permeation efficiency of the entrapped drug can be achieved by transfersomes [11]. Therefore, the obtained permeation rate of AA could be mainly attributed to the permeation of AA as loaded in transfersomes, and subsequent release of the AA substance.

Moreover, when compared to SP80AATG, the lower dynamic viscosity of TW80AATG and SDCAATG may also facilitate the permeation of the respective transfersomes. Furthermore, according to the HLB values of SP80 (4.3), TW80 (15) and SDC (16.7), SP80 has the highest affinity to lipophilic drug because of its relatively high lipophilicity. Since AA itself is lipophilic in nature, AA in the formulation may have an affinity toward the lipophilic SP80, resulting in retarded drug release of SP80AATG.

At 8 h, the percentage of AA penetration and flux exhibited by TW80AATG (8.53 ± 1.42% and 0.024 ± 0.008 mg/cm^2^/h, respectively) were not significantly different form SP80AATG (8.00 ± 1.70% and 0.019 ± 0.010 mg /cm^2^/h, respectively) but significantly higher than those of the SDCAATG (4.80 ± 0.50% and 0.014 ± 0.004 mg/cm^2^/h, respectively) and the control gels (*p* < 0.05).The higher percentage amount of AA within Strat-M^®^ membrane from TW80AATG after 8h compared to SDCAATG, is probably due to prolonged release of AA together with the penetration enhancing effect of TW80. In the present study, the relatively high amount of drug found in the TW80AATG (0.280 ± 0.005 g) and SP80AATG (0.279 ± 0.001 g) in comparison with the SDCAATG (0.272 ± 0.006 g) was found to enhance drug permeation and prolong drug release. According to the work presented by Tiwari et al. (2020), the non-ionic EAs such as TW80 could contribute to deformation of the transfersomes, thereby played a main role in improving the drug permeation [44]. Moreover, the concentration of the drug and the physicochemical characteristics of the formulation can also significantly affect the drug permeation [45,46]. The significantly higher penetration efficiency exhibited by AATGs compared to hydroethanolic AA in gel could be due to the constituents of transfersomes and this proves the requirement of a carrier base delivery system for AA dermal delivery. TW80AATG was selected as the optimal gel formulation based on the highest percentage of AA penetration and flux into the Strat-M^®^ membrane observed following 4 and 8 h of the application. 

### 2.2. Stability of AA Transfersomal Gels

The stability study of transfersomal gels (AATGs) was performed at three different temperature conditions, 4 ± 2 °C, 25 ± 2 °C and 40 ± 2 °C for 3 months. The physical appearance, AA content, dynamic viscosity and pH of AATGs were evaluated after 1 day and 1, 2 and 3 months of storage. There was no significant change observed in the physical appearance of the AATG over 3 months at any temperature condition (Figure 4).

At 4 °C, the AA content of SP80AATG, SDCAATG and TW80AATG significantly decreased at the end of 1, 2 and 3 months, respectively. At 25 °C, a significant decrease in the AA content was observed with both SP80AATG and TW80AATG at the end of 1 and 3 months, respectively. At 40 °C, only the AA content of SP80AATG significantly decreased after 3 months (*p* < 0.05) (Figure 5).

Generally, higher temperatures may result in a decrease of the drug content of a formulation. However, TW80AATG and SDCAATG did not exhibit any decrease in the AA content after storage at 40 °C for 3 months. According to the literature, the hydrophilic edge activator could cover the surface of vesicles more because of their larger hydrophilic moiety, which leads to a reduction in interfacial tension of the vesicles and a subsequent improvement in vesicular integrity [47]. Therefore, TW80 and SDC may facilitate the stability of the formulation because of their higher HLB values. However, the obtained AA content of all AATG formulations after storage at 4 °C, 25 °C and 40 °C for 3 months were still greater than 80% of initial AA content. Usually, based on the United States Pharmacopeia (USP) standards, the remaining drug content of the formulation in the range of 90–110% of the initial drug content is considered acceptable [48]. The observed AA content greater than 80% after 3 months may, however, indicate the compromised AA stability within these formulations. According to Surini et al. (2020), the stability of recombinant human epidermal growth factor (rhEGF) in transfersomal emulgel was reported to meet the criterion of 80–120% of drug content [49]. Moreover, in previous literature, this range of drug content was accepted and considered a reasonably good stability characteristic for transfersome gel [50]. Therefore, the lower limit of drug content in a transfersome-based formulation may be considerably not lower than 80% during shelf life. Nevertheless, further studies to check observed phenomenon are highly warranted. 

The dynamic viscosities of all AATG formulations stored at 4 °C were stable over 3 months of storage (Figure 6B). At 25 °C, the dynamic viscosity of SP80AATG significantly decreased at the end of 1 month, whereas that of SDCAATG significantly increased at the end of 1 month. Only TW80AATG had no change in dynamic viscosity over 3 months (Figure 6A). At 40 °C, the dynamic viscosity of TW80AATG was significantly increased at the end of 3 months, whereas those of SP80AATG and SDCAATG did not change during the study (Figure 6C). 

The increased dynamic viscosity of SDCAATG stored at 25 °C and TW80AATG stored at 40 °C may retard the movement and the fusion of the transfersomal vesicles, thereby improving the stability of the formulation [27,51]. However, it is well-known that increasing the viscosity of a formulation can decrease the skin permeation of the drug from an applied formulation. Therefore, low temperature conditions (at 4 °C and 25 °C), which could minimize the change in dynamic viscosity of transfersomal gel, were much preferable for AATG storage. Even though the AA content in the formulation was decreased under these storage conditions (at 4 °C and 25 °C), the remaining AA contents in the formulation were still greater than 80%.

When considering the pH of the AATGs stored at 4 °C, only TW80AATG showed a decrease in pH after 2 months (Figure 6E). At 25 °C, the pH of SDCAATG reduced at the end of 3 months, whereas that of SP80AATG increased after 1 month of storage. The TW80AATG showed no significant change in pH over 3 months (Figure 6D). At 40 °C, the pH of SDCAATG and TW80AATG decreased at the end of 1 and 2 months, respectively, while SP80AATG showed an increase in pH at the end of 1 month (*p* < 0.05) (Figure 6F). However, the obtained pH values of all AATG formulations at any storage condition after 3 months were within the acceptable range (5.3–7.6).

Generally, the integrity of transfersomes can be determined by different methods such as deformability measurement, stability study and microscopic examination by transmission electron microscope (TEM) [52,53] and scanning electron microscopy (SEM) [54]. However, according to the previous research studies related to the development and evaluation of transfersomal gel formulation, there were no specific test mentioned to determine the integrity of transfersomes after the incorporation in a gel base. A general conclusion with regard to the transfersomal integrity in a gel could be achieved by evaluating stability parameters such as drug content in the obtaining transfersomal gel formulations and the gel microscale structure examination by SEM [29,53]. In the present study, all transfersomal gel formulations exhibited the acceptable physical appearance, AA content, viscosity, pH and homogeneity. Hence, the integrity of the AA-loaded transfersomes was not majorly violated by the incorporation of vesicles into the gel formulation. Nevertheless, it is worth mentioning that the variation of physicochemical characteristics of the transfersomal gels over time may mediate outcomes in in vivo efficacy studies. Hence, further clinical investigation of AATG formulations is warranted to confirm these pre-clinical findings. 

In summary, a transfersomal gel exhibited promising in vitro dermal penetration of AA, which could be used as therapeutic approach against many types of skin disorders.

## 3. Materials and Methods

### 3.1. Materials

Asiatic acid (95%) was obtained from SEPPIC, Paris, France. Sodium deoxycholate (≥98%), sorbitan monooleate (Span 80) and polysorbate 80 (Tween 80) were purchased from SAFC^®^, Sigma-Aldrich, Buchs, Switzerland, HIMedia^®^ Laboratories Pvt. Ltd., Mumbai, India, and PanReac AppliChem, Darmstadt, Germany, respectively. L-α-lecithin soybean (purity > 94% phosphatidylcholine and <2% triglycerides) was obtained from EMD Millipore Corp., 290 Concord Rd. Billerica MA, USA, an affiliate of Merck KGaA, Darmstadt, Germany. Ethanol was bought from VWR International S.A.S, Briare, France. Hydroxyethyl acrylate/sodium acryloyldimethyl taurate copolymer (SEPINOV™ EMT 10), polyacrylate crosspolymer-6 (SEPIMAX™ ZEN) and phenoxyethanol were purchased from CHEMICO Inter Corporation CO., Ltd., Klongchan, Bangkapi, Bangkok, Thailand. Silicone-based emulsifier (ABIL CARE XL 80 MB) and hyaluronic acid (HYACARE FILLER CL) were purchased from Evonik Industries AG, ZI-Techasia Solutions Ltd., Klongtoey, Bangkok, Thailand. Acetonitrile (99.9%), ortho-phosphoric acid (85%), ethanol (99.9%) and methanol (99.9%) were purchased from RCI Labscan Co., Ltd., Pathumwan, Bangkok, Thailand. All other chemicals used were of analytical grade.

### 3.2. Methods 

#### 3.2.1. Preparation of AA Entrapped Transfersomes (AAT)

The transfersomes were prepared by a high-pressure homogenization technique as previously described [55,56]. Briefly, the total amount of 4% *w*/*w* of soybean lecithin and three different edge activators (EAs) including Tween 80 (TW80AAT), Span 80 (SP80AAT) and sodium deoxycholate (SDCAAT) at the ratio of 50:50, 90:10 and 90:10, respectively, was dissolved uniformly in ethanol to make lecithin-EA mixtures. Subsequently, 0.3% *w*/*w* of AA was dissolved in ethanol and gradually added into the lecithin-EA mixtures. The total amount of ethanol used for the formulation was 5% (*w*/*w*). Then, ultrapure water was added to top-up the mixture to 100 g. The resultant mixture was then homogenized using the Ultra Turrax^®^ homogenizer (T25 digital, IKA, Germany) at a speed of 10,000 rpm for 5 min at 60 °C. The mixture was then subjected to high pressure homogenization at 2000 bar for three cycles using the Microfluidizer LM20-30 (LM20-0068, PLC Holding Co., Ltd., Bangkok, Thailand).

#### 3.2.2. Preparation of AA Transfersomal Gels (AATGs)

The weighed amount of the gelling agents namely; hydroxyethyl acrylate/sodium acryloyldimethyl taurate copolymer (1% *w*/*w*) and polyacrylate crosspolymer-6 (1.5% *w*/*w*), preservative (0.8% *w*/*w*), silicone-based emulsifier (1% *w*/*w*) and hyaluronic acid (0.5% *w*/*w*) were dispersed in 95.2% *w*/*w* of AAT formulations (TW80AAT, SP80AAT and SDCAAT) with the homogenizing speed of 3000 rpm (Homogenize Mixer, Model RS-HGM 15720, Namsiang Group, Bangkok, Thailand) until plain gel is formed (TW80AATG, SP80AATG and SDCAATG). These amounts of gelling agents were selected based on former experiment to give the prepared gels a reasonable viscosity and homogeneity. Silicone-based emulsifier and hyaluronic acid were used to improve spreading characteristic of the product. The AATG formulations were continuously homogenized for a total of 30 min to obtain the homogeneous gels. Finally, the prepared AATG formulations were poured into glass bottles fitted with HDPE (high density polyethylene) closures and kept for stability studies under 25 ± 2 °C, 4 ± 2 °C and 40 ± 2 °C [57]. 

#### 3.2.3. Evaluation of AA Transfersomal Gels (AATGs)

##### Appearance

The appearance of the AATGs was evaluated by visual observation and taking a photo of a gel sample (1–2 g) on a watch glass after storage at 25 ± 2 °C, 4 ± 2 °C and 40 ± 2 °C for 1 day and 1, 2 and 3 months.

##### AA Content Determination

AA content in the transfersomal gels was determined by extracting AA from 1 g of gel. The gel formulation was placed in a 50 mL volumetric flask and topped up with methanol. The mixture was vortexed and sonicated at 35 °C in a water bath for 45 min to obtain an appropriate dissolution of the gel. Thereby, 2 mL of the resulting solution was diluted in a 10 mL volumetric flask using methanol as the solvent. This solution was filtered through a 0.45 µm syringe filter and AA content was determined using HPLC method. Agilent 1290 Infinity IILC (DEBAX00915, Agilent Technologies, Santa Clara, CA, USA) consisting of liquid chromatography pump (1290 flexible pump, G7104A, Serial No. DEBAX00915), UV–VIS detector (1290 DAD FS, G7117A, Serial No. DEBAV00877) and auto sampler (1290 vial sampler, G7129B, Serial No. DEBA900566) with the Luna^®^ 5µm RP-C18, 150 × 4.6 mm (H20-216135, 5291-0191, Phenomenex^®^) column were used to quantify the AA substance. The isocratic mobile phase was composed of acetonitrile (ACN): 0.05% orthophosphoric acid in water (45:55% *v*/*v*) ratio. Flow rate was kept constant at 1 mL/min and the injection sample volume was 20 µL. Column temperature was maintained at 25 °C and the eluent was detected at a wavelength of 206 nm by the diode array detector. Overall run time was set at 14 min [58,59,60]. The concentration of AA was estimated from the regression equation of the AA calibration curve [10].

##### pH Measurement

pH of the AATGs was measured by directly placing a pH meter (FiveEasy Plus, InLab Viscous Pro-ISM, Mettler Toledo Thailand Ltd., Bangkok, Thailand) electrode in contact with the gel. The measurement was performed three times and the mean ± SD was calculated.

##### Viscosity Measurement

Dynamic viscosity measurements were carried out at three different temperature conditions: 25 ± 0.1 °C, 4 ± 0.1 °C and 40 ± 0.1 °C using a rheometer (RheoStress 1, Thermo Haake, Germany) with a data logging software (Haake RheoWin 4.82.0002 Data Manager). Briefly, the AAT gel samples were allowed to rest for 300 s prior to analysis. The plate rotor was lowered until the gap between the rotor plate and the lower plate was 1.00 mm. The shear stress vs shear rate curve with a shear rate range of 0.1 to 1000 s^−1^ was obtained. It was used to determine the prospective dynamic viscosities [61,62]. 

##### Homogeneity Test

The homogeneity of the AATG formulations is important for the patient compliance following its application. The AATGs were tested for their homogeneity through two steps as follows; first, by visual inspection of the gels after set in the containers to determine the visual appearance and to detect any visible aggregates or particles present in the gels. Second, the AATGs were compared with Dermatix Ultra™, a commercially available scar-relieving gel, by separately pressing a small amount of both gels between the thumb and the index finger. Thereby, the consistency of the AATGs was noticed as homogeneous or not. For this purpose, 10 volunteer participants were asked to grade the AATG according to a grading system, which was allocated as (+++) good, (++) fair and (+) poor [10,16,57,63].

##### In Vitro Permeation Study

The in vitro permeation study of the AATG formulations were carried out using the Franz diffusion cell. Strat-M^®^ membranes were used as hypothetical skin during this test. This membrane was developed to mimic the layered structure and chemical features of human skin. It comprises a tight top layer supported by two layers of porous polyether sulfone (PES) on top of one single layer of polyolefin support, which resemble the epidermis and dermis layers. The porous membrane is coated with a combination of synthetic lipids, which impart a hydrophobic character to human stratum corneum. Strat-M^®^ membrane has various advantages such as low batch-to-batch variability, lack of storage limitations, safety and the ability to provide more consistent data [64]. A good correlation coefficient for the permeation of various drugs between Strat-M^®^ membrane and human skin was established [64,65]. Therefore, Strat-M^®^ membrane has been widely used for initial screening studies in the dermal permeation of topical drug delivery and formulations. However, there are some limitations existing in the utilizing of this artificial membrane. It was reported that the application of polyols, typical solvent used in topical formulations, appeared to disrupt the integrity of the Strat-M^®^ barrier. Hence, the utilizing of synthetic Strat-M^®^ membrane in the assessment of dermal permeation may be limited to the formulations with low polyol content [65]. Moreover, assessing the amount of the drug that penetrated into the viable epidermis/dermis layers cannot be estimated by using Strat-M^®^ because of the difficulty of separating the Strat-M^®^ membrane into layers [66]. Generally, a permeation study can be conducted in two approaches according to the dose of tested formulations applied to the skin: an infinite or a finite dose regimen. Each approach has its own advantages. For an infinite dose regimen, a large amount of the tested formulation is applied to the skin or membrane. Therefore, the dose could be considered constant throughout the study, leading to data that provides information about pharmacokinetic parameters. In a finite dose regimen, only a small amount of the tested formulation is applied to the skin, which best represents the in vivo situation. When it is necessary to know the real amount of drug absorbed into the different layers of the skin, the finite doses are required. However, to compare the penetration of various formulations through the skin, infinite dosage is chosen to evaluate the amount of drug permeated in the receptor fluid [67,68,69,70]. Therefore, in the present study, 1 g of tested gels as an infinite dose regimen was used for the in vitro permeation study to ensure that detectable amounts of drug in the receptor fluid will be obtained.

Briefly, these Strat-M^®^ membranes were mounted horizontally between the two compartments (donor and receptor) of the Franz diffusion cells. Each cell provides an effective diffusion area of 1.13 cm^2^. The receptor compartments were filled with 7.1 mL of a mixture of phosphate buffer saline (PBS) pH 7.4 and absolute ethanol (95:5, *v*/*v*). The receptor fluid was stirred continuously by a magnetic bar at 300 rpm and the temperature was maintained at 37 ± 2 °C. The prepared transfersomal gel formulation (1 g) was gently added into the donor compartment. Following that, the test was conducted without covering the top of the diffusion cells and thereby achieved a non-occlusive condition. Each of the receptor fluid volumes were taken out at predetermined time intervals (0.5, 1, 2, 4, 6 and 8 h). Those fluid samples were then frozen in a refrigerator and freeze dried (lyophilized) using a lyophilizer [71]. The resultant powder was dissolved in methanol and subjected to HPLC analysis to determine the amount of asiatic acid in the receptor fluid (AAR). Parallel to that, the remaining amount of the gel formulation in the donor compartment was appropriately taken out, diluted with methanol and vortexed vigorously. The resultant was filtered and followed by HPLC analysis to determine the remaining AA amount in the donor site (AAD) [56]. The amount of AA in the Strat-M^®^ membrane and the percentage of AA that penetrated into the Strat-M^®^ membrane, as well as the AA penetration rate or flux (mg/cm^2^/h) of the transfersomal gel formulations within the time period of 8 h, were determined by the following equations:AA amount in the membrane = Total AA − (AAR + AAD),(1)
%AA penetrated into the Strat-M^®^ membrane = [(Cumulative AA amount/Effective diffusion area)/Total AA amount] × 100(2)
Flux = (Cumulative AA amount/Effective diffusion area)/time(3)

Hydroethanolic solution of AA and physically mixed AA dispersions incorporated in gel were used as controls. Hydroethanolic solution of AA (HSAA) was prepared by dissolving a specific quantity (0.3%) of AA in 5% ethanol. Then, the resulting solution was topped-up with ultrapure water to 100 g and stirred using a magnetic stirrer for 5 min, then mixed using a homogenizer for 15 min. Physically mixed AA dispersions (PMDAA) were prepared using the same quantity of AA, lecithin, EA (TW80, SP80, SDC), ethanol and ultrapure water as in the transfersomal formulations. The resulting dispersion were then physically mixed using a glass rod. Finally, the gelling agents were then dispersed in HSAA and PMDAA to form the control gel formulations, which were subjected to the same in vitro permeation study.

#### 3.2.4. Stability of AA Transfersomal Gels

The transfersomal gel samples were filled into glass bottles fitted with HDPE closures and stored at three different temperature conditions, 25 ± 2 °C, 4 ± 2 °C and 40 ± 2 °C, for 3 months. The stability of the gel formulations was evaluated by analyzing their physical appearance, pH, dynamic viscosity and the AA content after 24 h, 1, 2 and 3 months. Each formulation was studied in triplicate and the average value was obtained as the result [72].

## 4. Conclusions

A transfersomal gel formulation for dermal administration of AA (AATG) was successfully developed. All AATG formulations appeared to be white in color and their pH values were compatible with the skin. The in vitro permeation study revealed that TW80AATG, SP80AATG and SDCAATG significantly increased the percentage of AA penetration and flux into the Strat-M^®^ membrane. TW80AATG was selected as the optimal gel formulation based on the highest percentage of AA penetration and flux into the Strat-M^®^ membrane observed following 4 h (14.20 ± 0.47% and 0.079 ± 0.015 mg/cm^2^/h, respectively) and 8 h (8.53 ± 1.42% and 0.024 ± 0.008 mg/cm^2^/h, respectively) of the application. The delivery of AA using simple dispersion gels did not result in high penetration at the site of application. The obtained dynamic viscosity and pH values of all AATG after storage at 4 °C, 25 °C and 40 °C for 3 months were considered acceptable. The observed AA content greater than 80% after 3 months of storage may indicate the compromised AA stability within these formulations. Low-temperature conditions (at 4 °C and 25 °C) were suggested as suitable storage conditions for AATG. The TW80AATG might serve as a lead formulation for further development toward scar prevention. 

## Figures and Tables

**Figure 1 molecules-27-04865-f001:**
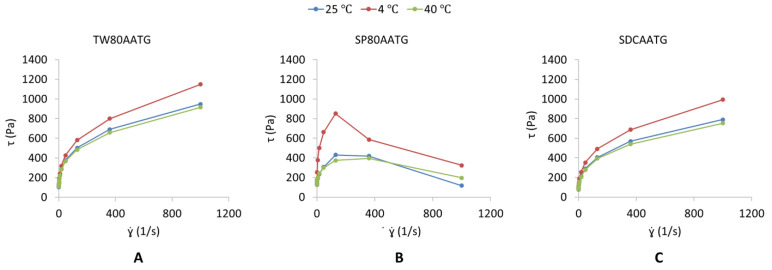
Shear stress [τ] (Pa) versus shear rate [$\dot{ɣ}$] (s^−1^) profile of (**A**) TW80AATG, (**B**) SP80AATG and (**C**) SCDAATG.

**Figure 2 molecules-27-04865-f002:**
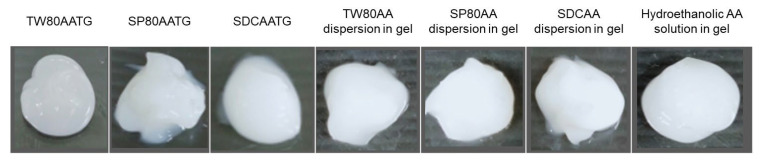
The physical appearance of freshly prepared AATGs and control gels for the in vitro permeation study.

**Figure 3 molecules-27-04865-f003:**
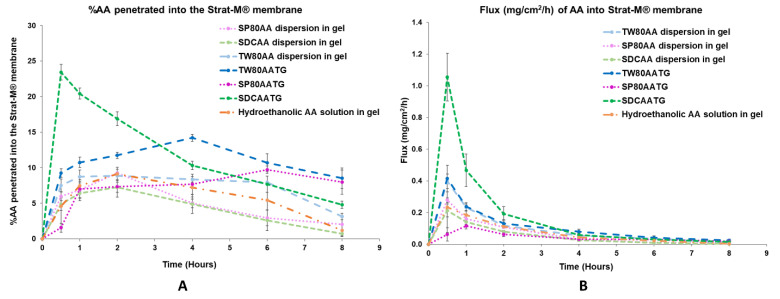
(**A**) In vitro permeation profile and (**B**) flux profile of the prepared AATGs and controls. Error bars indicate ± SD with *n* = 3.

**Figure 4 molecules-27-04865-f004:**
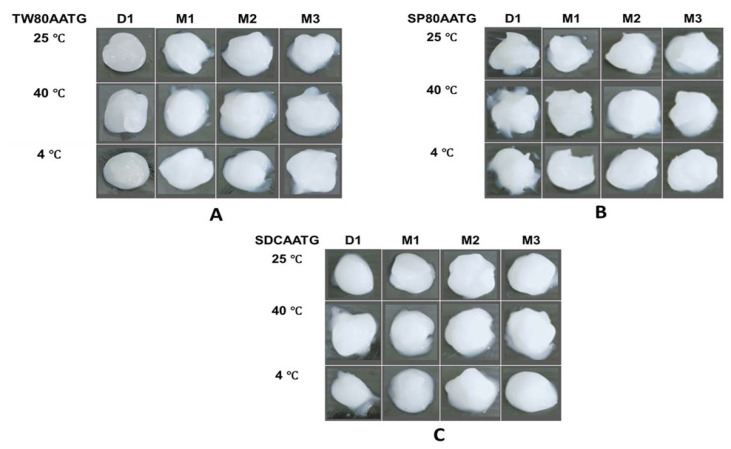
The physical appearance of AATGs during stability study period. (**A**) TW80AATG, (**B**) SP80AATG and (**C**) SDCAATG.

**Figure 5 molecules-27-04865-f005:**
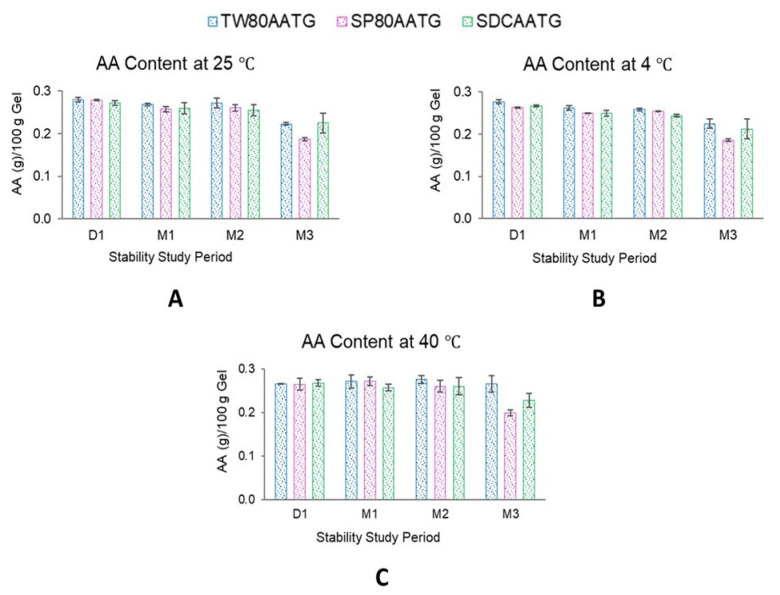
AA content of AATGs at different storage temperatures. (**A**) AA content at 25 °C, (**B**) AA content at 4 °C and (**C**) AA content at 40 °C. Each value represents mean ± SD with *n* = 3.

**Figure 6 molecules-27-04865-f006:**
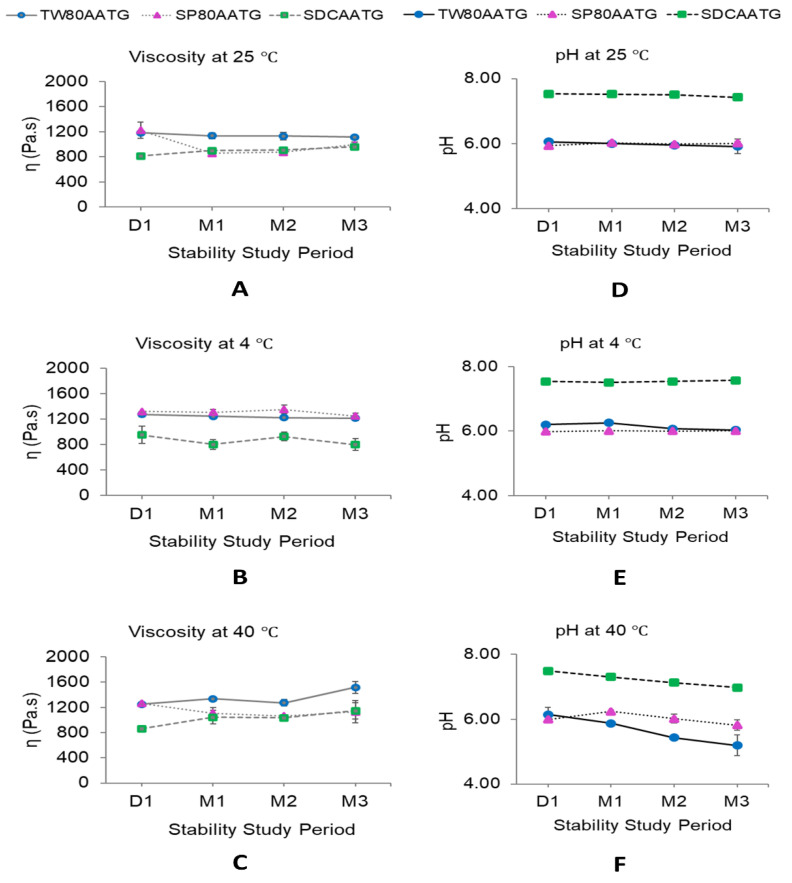
Dynamic viscosity and pH of AATGs during stability study period. (**A**) Viscosity at 25 °C, (**B**) viscosity at 4 °C, (**C**) viscosity at 40 °C, (**D**) pH at 25 °C, (**E**) pH at 4 °C and (**F**) pH at 40 °C. Each value represents mean ± SD with *n* = 3.

**Table 1 molecules-27-04865-t001:** The AA content, dynamic viscosity, pH and homogeneity of AA transfersomal gel formulations.

AATG	AA Content (AA g/100 g gel)	AA Content (%)	Dynamic Viscosity (Pa.s.)	pH	Homogeneity
TW80AATG	0.280 ± 0.005	97.85	1184.62 ± 6.45	6.06 ± 0.09	Good (+++)
SP80AATG	0.279 ± 0.001	97.69	1222.76 ± 131.99	5.94 ± 0.03	Good (+++)
SDCAATG	0.272 ± 0.006	95.26	812.21 ± 20.22	7.53 ± 0.03	Good (+++)

Results are presented as mean ± standard deviation (SD) with *n* = 3.

## Data Availability

Not applicable.

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
