# Peer review of "Preparation, Characterization and Permeation Study of Topical Gel Loaded with Transfersomes Containing Asiatic Acid"

_molecules, 2022, doi:10.3390/molecules27154865_

Round 1
Reviewer 1 Report
The article “Preparation, Characterization and Permeation Study of Topical Gel Loaded with Transfersomes Containing Asiatic Acid” by Opatha et al.
In my opinion the work is interesting however the paper is difficult to read, and the data discussion is quite poor.
Paper presentation needs to be deeply improved.
Furthermore, some major points have to be successfully addressed before publication.
Major points
- It could be useful to add a Table summarizing the tested formulations along with chemico-physical parameters.
- The gels seems to have quite a complex composition (hydroxyethyl acrylate/sodium 71 acryloyldimethyl taurate copolymer and polyacrylate crosspolymer-6, silicone-based emulsifier, hyaluronic acid and preservative) and the function of each component is not completely clear.
- Integrity of transferosomes in the gel formulation must to be evaluated.
- The AA release from transferosome and from transferosome/gel must be evaluated
- Lines 166-168
“The % permeation and flux of AA into the Strat-M® membrane over 8 hours period was plotted against the function of time to evaluate the release profiles (Figure 5).”
Lines 192-193
“…leading to increased permeation of the vesicles and subsequent AA release.”
The permeation rate of AA is reported. It is not clear if AA permeated as free substance or loaded in trasferomes.Please clarify.
- Line 328-329 HPLC method must be described in details
Reviewer 2 Report
The present manuscript deals with the ability of gel formulations containing asiatic acid (AA)-loaded transfersomes to overcome the typical drawbacks of transfersomes (predominately low viscosity) and thus to ensure the more convenient treatment of skin disorders. The authors already published data concerning physicochemical characterization of developed transferosomes as well as in vitro permeation and anti-inflammatory activity of AA (Opatha SAT et al. EJPB-D-22-00195, Available at SSRN: https://ssrn.com/abstract=4063752). However, there are several important remarks which limit the acceptance of this manuscript in the present form. My suggestions for authors to improve the quality of this manuscript are follows:
Introduction:
1. The authors emphasized that AA can possess many biological activities such as anti-skin cancer, scar prevention, anti-inflammatory, anti-microbial as well as significant antioxidant activities. However, it is important to emphasize the skin disorders that can be potentially treated with developed gels containing AA-loaded nano-transferosomes.
Materials and methods:
1. The manufacturer of soyabean lecithin should be added.
2. The transferosome preparation should be described in more details. Please add the amount of ethanol used to dissolve lecithin and for the preparation of AA solution. The amount of ethanol is important since it can affect skin barriers properties. Which preservative and silicone-based emulsifier were used? Please add the commercial name and manufacuturer of preservative, silicone based emulsifier and hyaluronic acid.
3. Which homogenizer was used for sample preparation?
4. Please explain why 1g of tested gels was used for in vitro permeation study.
Results and discussion:
1. Figure 1 (Schematic illustration of the preparation of AA transfersomal gels and subsequent determination of dermal permeation of asiatic acid using the respective transfersomal gel nanocarriers) could be useful as a graphical abstract. It does not contain any important results and therefore, it should be removed from the manuscript.
2. Figure 2 showing the physical appearance of gels containing AA-loaded transfersomes should be removed (the same pictures are the part of Figure 4).
3. The term “permeation” is usually used to describe drug amount the permeate through the skin (or one or more skin layers). Therefore, according to my opinion, instead % permeation of AA into the Strat-M® membrane over 8 hours period, should stand % of AA penetrated or retained into the Strat-M® membrane. In other words, the term permeation can be used only for amount of AA in the receptor compartment.
4. The authors presented the %amount of AA in the membrane as a function of time. As authors emphasized, the amount of AA in membrane was calculated as difference of total amount of AA and amount recovered from the donor and receptor compartment. For the SDCAATG formulation, the highest amount of AA within the Strat-M® membrane was achieved after 0,5 h and declined over the rest of test, probably due to diffusion of AA in the receptor compartment. Therefore, it could be interesting to add the graph showing the relation of the cumulative amount of AA permeated per unit area (mass unit/cm2) in receptor compartment as function of time for each tested formulation. Obviously, there are significant differences in the AA release kinetics among the tested formulations.
The higher % amount of AA within Strat-M® membrane from TW80AATG after 8h compared to SDCAATG, is probably due to prolonged release of AA, rather than to penetration enhancing effect of TW80.
5. The authors selected TW80AATG formulation as the optimal based on the highest %amount of AA retained in membrane and flux following 4 hours and 8 hours of the application. My suggestion is to analyze separately amount of AA in the receptor and donor compartment, maybe some formulation is more suitable to deliver AA to membrane/skin layers and other to underlying tissue or systemic circulation.
6. Also, limitations and advantages of utilizing synthetic Strat-M® membrane have to be clearly stated in the manuscript.
5. “At 4 °C, the AA content of SP80AATG, SDCAATG, and TW80AATG significantly decreased at the end of 1, 2 and 3 months, respectively”. This is quite unusual, as generally the lower storage temperature can contribute to maintain drug chemical stability. What can be possible explanation for observed phenomenon?
6. “However, the obtained AA contents of all AATG formulations at any storage condition after 2 months were still higher than 90 % of initial drug content [37] and only TW80AATG and SDCAATG exhibited more than 80 % of AA content from the initial drug content after 3 months under any storage condition.” This sentence should be rephrased to make a point clearer.
7. The authors wrote: “Therefore, low temperature conditions (at 4 °C and 25 °C), which could minimize the change in dynamic viscosity of transfersomal gel, were much preferable for the AATG storage.” However, this storage conditions are not appropriate regrading AA content.
8. According to my opinion, the monitoring the stability of transfersomal gel during the 3 months is not sufficient to make any valid conclusion (particularly, due to observed differences in selected physicochemical parameters). It can not be claimed at this stage of product development that TW80AATG formulation can be applied for further clinical investigation.
Round 2
Reviewer 1 Report
Accept in the current form
Author Response
Thank you very much for your kind consideration.
Reviewer 2 Report
I would like to thank to authors for detailed answers on my comments. I want only to add that AA content greater than 80% after 3 months can not be completely acceptable (usually 90-110% is acceptable). According to my opinion, this may indicate the compromised AA stability within these formulations. Further studies are required to check observed phenomenon. The manuscript can be accepted after correction of this small remark.
